# Risk factors for mortality of coronavirus disease-2019 (COVID-19) patients in two centers of Hubei province, China: A retrospective analysis

**Xiao-Bin Zhang** [1,2☯] *, **Lan Hu** [3☯], **Quan Ming** [4☯], **Xiao-Jie Wei** [5☯], **Zhen-Yu Zhang** [6☯], **Li-Da Chen** [7☯], **Ming-Hui Wang** [1,2☯], **Weng-Zhen Yao** [1,2☯], **Qiu-Fen Huang** [1,2☯], **Zhang-Qiang Ye** [1,2☯], **Yu-Qing Cai** [1,2☯], **Hui-Qing Zeng** [1,2☯] *

1 Department of Pulmonary and Critical Care Medicine, Zhongshan Hospital, Xiamen University, Fujian, China, 2 Teaching Hospital of Fujian Medical University, Fujian, China, 3 Department of Gastroenterology, Optic Valley division of Tongji Hospital, Tongji Medical College, Huazhong University of Science and Technology, Wuhan, China, 4 Yichang Third People's Hospital, Third People's Hospital Affiliated to SanXia University, China, 5 Department of Pulmonary and Critical Care Medicine, Third People's Hospital Affiliated to Fujian University of Traditional Chinese Medicine, China, 6 Department of Geriatrics, Zhongshan Hospital, Xiamen University, Fujian, China, 7 Department of Pulmonary and Critical Care Medicine, Zhangzhou Hospital Affiliated to Fujian Medical University, Fujian, China

☯ These authors contributed equally to this work.
* zhangxiaobincn@xmu.edu.cn (XBZ); zhq20071212@xmu.edu.cn (HQZ)

**Data Availability Statement:** All relevant data are within the manuscript and its Supporting Information files.

## Abstract

### Purpose

Since the outbreak in late December 2019 in Wuhan, China, coronavirus disease-2019 (COVID-19) has become a global pandemic. We analyzed and compared the clinical, laboratory, and radiological characteristics between survivors and non-survivors and identify risk factors for mortality.

### Methods

Clinical and laboratory variables, radiological features, treatment approach, and complications were retrospectively collected in two centers of Hubei province, China. Cox regression analysis was conducted to identify the risk factors for mortality.

### Results

A total of 432 patients were enrolled, and the median patient age was 54 years. The overall mortality rate was 5.09% (22/432). As compared with the survivor group (n = 410), those in the non-survivor group (n = 22) were older, and they had a higher frequency of comorbidities and were more prone to suffer from dyspnea. Several abnormal laboratory variables indicated that acute cardiac injury, hepatic damage, and acute renal insufficiency were detected in the non-survivor group. Non-surviving patients also had a high computed tomography (CT) score and higher rate of consolidation. The most common complication causing death was acute respiratory distress syndrome (ARDS) (18/22, 81.8%). Multivariate Cox

**Funding:** This work was supported by Grant 2018-2-65 for Youth Research Fund from Fujian Provincial Health Bureau, Grant 2020GGB057 for Young people training project from Fujian Province Health Bureau, and Grant 2018J01393 for Fund from Natural Science Foundation of Fujian Province, China.

**Competing interests:** The authors have declared that no competing interests exist.

regression analysis revealed that hemoglobin (Hb) <90 g/L (hazard ratio, 10.776; 95% confidence interval, 3.075–37.766; $p<0.0001$), creatine kinase (CK-MB) >8 U/L (9.155; 2.424–34.584; $p = 0.001$), lactate dehydrogenase (LDH) >245 U/L (5.963; 2.029–17.529; $p = 0.001$), procalcitonin (PCT) >0.5 ng/ml (7.080; 1.671–29.992; $p = 0.008$), and CT score >10 (39.503; 12.430–125.539; $p<0.0001$) were independent risk factors for the mortality of COVID-19.

## Conclusions

Low Hb, high LDH, PCT, and CT score on admission were the predictors for mortality and could assist clinicians in early identification of poor prognosis among COVID-19 patients.

## Introduction

In December 2019, coronavirus disease-2019 (COVID-19) was caused by the severe acute respiratory syndrome coronavirus 2 (SARS-CoV-2) outbreak in Wuhan, Hubei Province, China [1]. Subsequently, it rapidly spread all over the world and became a global pandemic. As of April 29, 2020, more than 3 million COVID-19 cases have been reported worldwide, causing more than 200,000 deaths. This pandemic started from zoonotic transmission, but human-to-human transmission was soon confirmed [2]. The clinical spectrum of COVID-19 varies from asymptomatic, mild upper airway illness, to severe pneumonia with respiratory failure. Common symptoms of COVID-19 include fever, cough, fatigue, and dyspnea [3]. Being a beta coronavirus, the SARS-CoV-2 virus was closely related (with 88% identity) to two bat-derived severe acute respiratory syndrome (SARS)-like coronaviruses, bat-SL-CoVZC45 and bat-SL-CoVZXC21, collected in 2018 in Zhoushan, eastern China, but were more distant from SARS-CoV (about 79%) and MERS-CoV (about 50%) [4].

Evidence shows that the overall mortality rate of COVID-19 is 3.77–5.4% [5–7], however, it increases up to 41.1–61.5% among severe or critically ill patients [8–10]. To reduce the overall mortality rate, identifying the risk factors related to disease severity and mortality in COVID-19 patients is urgently required. Previous studies have shown that older age, underlying comorbidities, high D-dimer level, and abnormalities of several biochemical variables were closely associated with disease severity or even death of COVID-19 patients [6, 9, 11–13]. Most of the previous studies were single-center investigations with small sample sizes in Wuhan, China. The present study analyzed and compared the demographic, clinical, and laboratory variables, and radiological features between surviving and no-surviving patients with laboratory-confirmed COVID-19 in two hospitals with a relatively larger sample size. Potential risk factors for death on admission were determined. We tried to provide some useful information to predict the death of COVID-19 patients through this retrospective cohort study of 432 cases in two centers in Hubei province, China.

## Methods

### Study population

Adult inpatients (≥18 years old) with COVID-19 were retrospectively analyzed from the following two centers: Optic Valley division of Tongji Hospital, Tongji Medical College, Huazhong University of Science and Technology, and Wuhan and Yichang Third People's Hospital, Hubei Province from January 20, 2020, to March 30, 2020. All patients were treated

by the Fujian and Xiamen Medical Team that reached to help Hubei province. COVID-19 diagnosis was based on the New Coronavirus Pneumonia Prevention and Control Program published by the Chinese National Health Commission (version 6) [14]. All patients were laboratory-confirmed positive cases of SARS-CoV-2 by quantitative reverse transcriptase-polymerase chain reaction (qRT-PCR) of nasopharyngeal swab samples and had a definite outcome (dead or discharged). The study was approved by the Institutional Ethics Committee of both Optic Valley division of Tongji Hospital, Tongji Medical College, Huazhong University of Science and Technology, and Wuhan and Yichang Third People's Hospital.

## Data collection

All clinical information, including age, sex, medical history, comorbidities, laboratory findings, and thoracic computed tomography (CT) results, treatment approaches, and complications, of COVID-19 patients who had been discharged from or had died at the two centers was extracted from electronic medical records. All data were collected using an electronic data collection form. In order to verify the accuracy, two researchers independently reviewed the data collection forms.

## Definition

The discharge criteria were absence of fever for 3 days, substantial improvement in both lungs on thoracic CT, clinical remission of respiratory symptoms, and two consecutive nasopharyngeal swab samples negative for SARS-CoV-2 RNA obtained at least 24 hours apart. COVID-19 severity was defined according to the New Coronavirus Pneumonia Prevention and Control Program published by the Chinese National Health Commission (version 6). Acute respiratory distress syndrome (ARDS) was diagnosed according to the Berlin definition [15].

## Laboratory testing

Nasopharyngeal swab samples were collected for SARS-CoV-2 viral nucleic acid detection using real-time reverse transcriptase-polymerase chain reaction (RT-PCR). Blood was obtained from all patients on admission for analysis. The BC 3000 auto hematology analyzer (Mindray Medical International, Inc., Shenzhen, China) was used for routine blood tests. Serum renal and liver function, creatine kinase (CK), CK-MB, lactate dehydrogenase (LDH), C-reactive protein (CRP), procalcitonin (PCT), and erythrocyte sedimentation rate (ESR) were measured on a Beckman Coulter AU5800 (Beckman Coulter Co, Brea, CA, USA). Lymphocyte subsets were analyzed on a BD FACS Canto II flow cytometry system (BD Biosciences, CA, USA). Blood coagulation profiles were analyzed by immunoturbidimetry using the ACL TOP system (Intrstumentation Laboratory, Milan, Italy). Arterial blood gas analysis was performed with a hamo gas analyzer (Nova Biomedical, USA). The levels of serum cytokines [Interleukin-1 (IL-1), IL-2 receptor, IL-6, IL-8, IL-10, and tumor necrosis factor-⟨ (TNF-⟨)] were measured by the chemilumnescent immunoassay (CLIA) on Siemens Immulite 1000 analyzer according to the manufacturer's instructions.

## CT image acquisition and scoring

A thoracic CT scan was performed before or after 2 days of admission in all patients. Two researchers reviewed and scored the thoracic CT images independently with the same CT score criteria: lobar involvement was classified as 0-none (0%), 1-minimal (1–25%), 2-mild (26–50%), 3-moderate (51–75%), or 4-severe (76–100%) of each lobe. Finally, a total score of 0–20 for the five lobes was summarized.

## Data statistical analysis

SPSS 22.0 software was used for statistical analysis. Continuous data are presented as means± standard deviation or median (interquartile range, IQR), while categorical data are presented as number (%). Means of continuous variables were compared using the *Mann-Whitney* test. Categorical variables were compared using the *chi-square* test or *Fisher's* exact test between the groups. Cox regression analysis was used to explore the risk factors for COVID-19 mortality. Univariate and multivariate regression models were used. A variable was entered into the multivariable Cox regression model when it satisfied one of the following items: 1. univariate Cox regression showed a significant difference; 2. Variable was closely associated with survival according to previous reports [6, 10–13] and clinical experience. If the proportion of missing data of a variable was less than 20%, the missing data were replaced by the mean value; however, when the proportion of missing data was more than 20%, the variable was not entered into Cox regression. A *p*-value of less than 0.05 was considered statistically significant.

## Results

### Clinical features on admission

As shown in Table 1, a total of 432 patients were included in the final analysis (128 from the Optic Valley division of Tongji Hospital, Tongji Medical College, Huazhong University of Science and Technology, Wuhan and 304 from Yichang Third People's Hospital). A total of 22 (5.09%) patients died during hospitalization and 410 (94.91%) patients were discharged. Male patients accounted for 53.2% of all patients. The median age was 54 years (IQR 39–66 years), and patients in the non-survivor group were much older than those in the survivor group [66 years (54–70 years) vs. 53 years (38–66 years), *p* = 0.003]. The comorbidity rate in the non-survivor group was higher than that in the survivor group (77.3% vs. 31.0%, *p*<0.0001). Higher rates of hypertension, cardiovascular or cerebrovascular diseases, diabetes, chronic kidney disease, and other diseases were observed in the non-survivor group than in the survivor group (all *p*≤0.001). The top five symptoms in all patients were fever (71.3%), dry cough (62.5%), sputum production (30.1%), fatigue (29.6%), and chest tightness (10.9%). We found that patients in the non-survivor group had significantly higher dyspnea rates than those in the survivor group (36.4% vs. 6.6%, *p*<0.0001). Patients in the non-survivor group had higher heart rate and respiratory rate than those in the survivor group (all *p*<0.05). The median number of days from the onset of illness to hospital admission was 5 days (IQR 2–10 days). The median time from illness onset to discharge was 26 days (IQR 20–36 days), whereas the median time to death was 18 days (IQR 10–26 days) (*p* = 0.001). Regarding the disease severity between groups, the proportions of severe type and critical type in the non-survivor group were significantly higher than those in the survivor group (72.7% vs. 24.6% for the severe type and 27.3% vs. 0.5% for the critical type, all *p*<0.0001).

### Laboratory findings

Compared with the results in the survivor group, lymphocyte count, CD4+ T cells, CD8+ T cells, hemoglobin, platelet count, albumin, arterial partial pressure of oxygen (PaO2), and the ratio of $PaO_2$ to the fraction of inspired $O_2$ ($FiO_2$) were significantly lower in the non-survivor group. The levels of variables reflecting liver function (total bilirubin, direct bilirubin, and aspartate aminotransferase), renal function (cystatin C), cardiac injury (creatine kinase-MB, cardiac troponin I, and brain natriuretic peptide), inflammation (CK, LDH, CRP, ESR, and

**Table 1. Comparison of demographic and clinical characteristics of COVID-19 patients between the survivor and non-survivor groups.**

| Variable | Total (n = 432) | Non-survivors (n = 22) | Survivors (n = 410) | p value |
|---|---|---|---|---|
| Age, years | 54 (39–66) | 66 (54–70) | 53 (38–66) | 0.003 |
| Gender, n (%) | | | | 0.754 |
| Male | 230 (53.2) | 11 (50.0) | 219 (53.4) | .. |
| Female | 202 (46.8) | 11 (50.0) | 191 (46.6) | .. |
| Comorbidity, n (%) | 144 (33.3) | 17 (77.3) | 127 (31.0) | <0.0001 |
| Hypertension | 97 (22.5) | 13 (59.1) | 84 (20.5) | <0.0001 |
| ACEI or ARB administration | 20 (20.6) | 3 (23.1) | 17 (20.2) | 0.814 |
| Cardiovascular or cerebrovascular diseases | 25 (5.8) | 7 (38.1) | 18 (4.4) | <0.0001 |
| Diabetes | 56 (13.0) | 8 (36.4) | 48 (11.7) | 0.001 |
| Chronic obstructive pulmonary disease | 25 (5.8) | 3 (13.6) | 22 (5.4) | 0.106 |
| Carcinoma | 5 (1.2) | 0 (0.0) | 5 (1.2) | 0.602 |
| Chronic kidney disease | 9 (2.1) | 5 (22.7) | 4 (1.0) | <0.0001 |
| Others | 25 (5.8) | 7 (31.8) | 18 (4.4) | <0.0001 |
| Symptom | | | | |
| Fever | 308 (71.3) | 15 (68.2) | 293 (71.5) | 0.740 |
| Dry cough | 270 (62.5) | 15 (68.2) | 255 (62.2) | 0.572 |
| Fatigue | 128 (29.6) | 7 (31.8) | 121 (29.5) | 0.818 |
| Dyspnea | 35 (8.1) | 8 (36.4) | 27 (6.6) | <0.0001 |
| Sputum production | 130 (30.1) | 6 (27.3) | 124 (30.2) | 0.767 |
| Sore throat | 34 (7.9) | 2 (9.1) | 32 (7.8) | 0.827 |
| Chest tightness | 47 (10.9) | 2 (9.1) | 45 (11.0) | 0.782 |
| Diarrhoea | 20 (4.6) | 1 (4.5) | 19 (4.6) | 0.985 |
| Myalgia | 46 (10.6) | 3 (13.6) | 43 (10.5) | 0.641 |
| Headache | 19 (4.4) | 1 (4.5) | 18 (4.4) | 0.972 |
| Temperature, ˚C | 36.8 (36.5–37.3) | 37.0 (36.7–37.6) | 36.8 (36.5–37.3) | 0.221 |
| Heart rate, beats/min | 88 (80–97) | 91 (85–101) | 88 (80–97) | 0.046 |
| Respiratory rate, breaths/min | 21 (20–23) | 22 (21–24) | 21 (20–22) | 0.016 |
| Systolic blood pressure, mmHg | 126 (118–137) | 129 (120–137) | 126 (118–138) | 0.941 |
| Diastolic blood pressure, mmHg | 80 (71–87) | 77 (67–81) | 80 (71–87) | 0.178 |
| Illness onset to hospital admission, days | 5 (2–10) | 5 (2–7) | 5 (2–10) | 0.482 |
| Illness onset to hospital discharge/death, days | 26 (20–36) | 18 (10–26) | 26 (20–36) | 0.001 |
| Disease severity status* | | | | |
| Mild | 16 (3.7) | 0 (0.0) | 16 (3.9) | 0.345 |
| Moderate | 291 (67.4) | 0 (0.0) | 291 (71.0) | <0.0001 |
| Severe | 117 (27.1) | 16 (72.7) | 101 (24.6) | <0.0001 |
| Critical | 8 (1.9) | 6 (27.3) | 2 (0.5) | <0.0001 |

Abbreviations: COVID-19: coronavirus disease-2019; ACEI: angiotensin converting enzyme inhibitors; ARB: angiotensin receptor blocker.

*: Classification of COVID-19 severity is according to the New Coronavirus Pneumonia Prevention and Control Program published by the Chinese National Health Commission (version 6).

PCT), cytokine levels [IL-2 receptor, IL-6, IL-8, and IL-10], and coagulation (D-dimer) were significantly higher in the non-survivor group as compared with the survivor group (all $p<0.05$) (Table 2). The other variables, including white blood cell count, alaine aminotransferase, creatinine, myoglobin, serum lactate, prothrombin time, activated partial thromboplastin time, and arterial partial pressure of carbon dioxide, were not different between the survivor and non-survivor groups (all $p>0.05$).

**Table 2. Comparison of laboratory findings of COVID-19 patients between the survivor and non-survivor groups.**

| Variable | Total (n = 432) | Non-survivors (n = 22) | Survivors (n = 410) | *p* value |
|---|---|---|---|---|
| White blood cell count, ×10$^9$/L | 4.70 (3.59–6.10) | 4.45 (3.15–8.98) | 4.70 (3.60–6.04) | 0.927 |
| Lymphocyte count, ×10$^9$/L | 1.21 (0.84–1.63) | 0.66 (0.53–0.89) | 1.23 (0.87–1.65) | <0.0001 |
| T cell subsets | | | | |
| CD4+ T cells, cell/μL | 593 (339–848) | 85 (68–125) | 657 (484–875) | <0.0001 |
| CD8+ T cells, cell/μL | 339 (217–490) | 125 (98–198) | 369 (257–515) | <0.0001 |
| Haemoglobin, g/L | 124 (112–136) | 113 (104–122) | 125 (113–137) | 0.001 |
| Platelet count, ×10$^9$/L | 158 (119–208) | 132 (80–169) | 160 (121–209) | 0.01 |
| Albumin, g/L | 38.2 (33.9–41.4) | 30.9 (28.0–37.4) | 38.5 (34.3–41.5) | <0.0001 |
| Total bilirubin, μmol/L | 9.30 (6.81–13.15) | 14.40 (8.60–20.00) | 9.23 (6.80–12.60) | 0.001 |
| Direct bilirubin, μmol/L | 2.71 (1.88–3.96) | 4.87 (2.44–7.41) | 2.68 (1.86–3.88) | 0.004 |
| Alaine aminotransferase, U/L | 22.0 (14.0–35.0) | 25.5 (14.0–50.0) | 22.0 (14.0–35.0) | 0.194 |
| Aspartate aminotransferase, U/L | 22.0 (16.0–30.0) | 36.0 (21.3–67.3) | 21.0 (16.0–29.0) | 0.001 |
| Creatinine, μmol/L | 68.5 (55.9–82.5) | 67.5 (62.8–92.3) | 68.5 (55.5–81.9) | 0.127 |
| Cystatin C, mg/L | 1.05 (0.89–1.29) | 1.31 (1.05–2.28) | 1.03 (0.88–1.23) | 0.002 |
| Creatine kinase-MB, U/L | 8.9 (1.3–12.2) | 14.2 (8.8–35.3) | 8.8 (1.2–11.4) | <0.0001 |
| Myoglobin, ng/mL | 36.0 (27.8–79.1) | 49.8 (30.2–125.3) | 35.0 (27.8–77.5) | 0.258 |
| Cardiac troponin I, pg/mL | 4.5 (1.9–11.1) | 13.0 (4.6–26.3) | 4.3 (1.9–10.3) | 0.001 |
| Brain natriuretic peptide, pg/mL | 104.5 (29.0–311.0) | 614.5 (304.6–1177.5) | 96.5 (25.3–244.5) | <0.0001 |
| Creatine kinase, U/L | 62.0 (43.0–98.0) | 117.0 (52.0–201.8) | 60.0 (42.3–94.8) | 0.04 |
| Lactate dehydrogenase, U/L | 206.0 (169.0–265.8) | 399.0 (235.0–595.0) | 203.5 (167.0–260.0) | <0.0001 |
| C reactive protein, mg/L | 12.5 (3.1–38.7) | 63.9 (24.7–101.6) | 11.7 (2.8–35.8) | <0.0001 |
| Erythrocyte sedimentation rate, mm/h | 25.0 (14.5–48.0) | 40.6 (23.0–55.5) | 24.0 (13.0–46.5) | 0.015 |
| Procalcitonin, ng/mL | 0.07 (0.05–0.11) | 0.17 (0.10–0.33) | 0.07 (0.05–0.10) | <0.0001 |
| Serum lactate, mmol/L | 1.84 (1.49–2.34) | 1.96 (1.49–2.46) | 1.81 (1.48–2.27) | 0.509 |
| Interleukin-2 receptor, U/mL | 413.0 (266.8–697.3) | 737.0 (697.0–840.6) | 380.0 (261.0–646.0) | 0.01 |
| Interleukin-6, pg/mL | 3.5 (1.6–11.9) | 165.0 (115.3–257.9) | 2.8 (1.5–6.7) | <0.0001 |
| Interleukin-8, pg/mL | 9.3 (6.3–12.9) | 23.5 (9.9–65.7) | 9.0 (6.0–12.1) | 0.003 |
| Interleukin-10, pg/mL | 5.0 (5.0–5.0)* | 6.4 (5.0–8.4) | 5.0 (5.0–5.0)* | 0.009 |
| TNF-⟨, pg/mL | 7.6 (6.1–9.7) | 8.3 (7.0–19.0) | 7.5 (6.0–9.7) | 0.110 |
| Prothrombin time, s | 11.1 (10.6–13.0) | 11.2 (10.7–12.0) | 11.1 (10.5–13.0) | 0.975 |
| Activated partial thromboplastin time, s | 32.3 (27.7–37.0) | 30.7 (24.0–36.1) | 32.4 (28.0–37.1) | 0.234 |
| D-dimer, μg/ml | 0.54 (0.44–0.80) | 1.46 (0.74–5.09) | 0.53 (0.43–0.70) | <0.0001 |
| PaO$_2$, mmHg | 92.0 (76.5–110.0) | 65.4 (56.1–77.8) | 94.1 (80.0–112.0) | <0.0001 |
| PaCO$_2$, mmHg | 40.4 (36.9–43.6) | 37.6 (29.9–47.1) | 40.5 (37.4–43.6) | 0.147 |
| PaO$_2$: FiO$_2$, mmHg | 306.9 (231.8–362.1) | 91.7 (80.5–196.5) | 316.1 (258.1–366.2) | <0.0001 |

*: minimum detection value is 5.0.

All data are presented as median interquartile (IQR).

The normal ranges of all laboratory findings are outlined in S5 Table.

Abbreviations: COVID-19: coronavirus disease-2019; TNF: tumor necrosis factor-⟨. PaO$_2$: arterial partial pressure of oxygen; PaCO$_2$: arterial partial pressure of carbon dioxide; FiO$_2$: fraction of inspired O$_2$.

## CT image results

The most common finding of the CT scan was ground-glass opacity (98.6%), followed by bilateral lung involvement (82.5%) and consolidation (23.3%). Compared with the survivor group, patients in the non-survivor group had a higher rate of consolidation (45.5% vs. 22.1%, *p* = 0.012) and increased CT score [14.0 (10.5–16.0) vs. 6.0 (4.0–7.3), *p*<0.0001] (Table 3).

**Table 3. Comparison of CT image results of COVID-19 patients between the survivor and non-survivor groups**[*].

| Variables | Total (n = 416) | Survivors (n = 394) | Non-survivors (n = 22) | p value |
|---|---|---|---|---|
| Bilateral lung involvement | 343 (82.5) | 322 (81.7) | 21 (95.5) | 0.099 |
| Ground-glass opacity | 410 (98.6) | 388 (98.5) | 22 (100.0) | 0.720 |
| Consolidation | 97 (23.3) | 87 (22.1) | 10 (45.5) | 0.012 |
| Pleural effusion | 5 (1.2) | 4 (1.0) | 1 (4.5) | 0.139 |
| Pleural thickening | 5 (1.2) | 5 (1.3) | 0 (0.0) | 0.761 |
| CT score | 6.0 (4.0–8.0) | 6.0 (4.0–7.3) | 14.0 (10.5–16.0) | <0.0001 |

[*]: moderate, severe, and critically ill cases were included in this analysis.

Abbreviations: CT: computed tomography; COVID-19: coronavirus disease-2019.

## Treatment and complications

Nearly all patients (96.5%) received antiviral drugs during hospitalization. There were no differences between the survivor and non-survivor groups in terms of usage of the antiviral drug (umifenovir, interferon 〈 nebulization, lopinavir/ritonavir, ribavirin, chloroquine, hydroxychloroquine, and oseltamivir). A total of 95.5% of patients in the non-survivor group and 87.1% of patients in the survivor group received antibiotic therapy ($p = 0.247$). There was a significant difference in corticosteroid and intravenous immunoglobulin usage between the two groups (all $p<0.0001$). As compared with the survivor group, more patients in the non-survivor group received non-invasive or invasive mechanical ventilation (all $p<0.0001$). The frequency of complications in the non-survivor group was noticeably higher than that in the survivor group. The most frequently detected complication in the non-survivor group was ARDS (81.8%), followed by acute hepatic insufficiency, heart failure, thrombocytopenia, acute kidney injury, and acute cardiac injury. The median time of viral shedding after COVID-19 onset was 6 days (IQR 3–10 days), and there was no difference in viral shedding time between the two groups (Table 4).

## Prediction of mortality

In order to evaluate the risk factors for hospital admission for mortality, Cox regression analysis was conducted. We initially performed univariate analysis by using the variables that were statistically significant (all $p<0.05$) between the non-survivor and survivor groups in Tables 1–3. The univariate analysis results (Table 5) showed that the following variables were associated with the mortality of COVID-19 patients: age $\geq$ 65 years, hypertension, cardiovascular or cerebrovascular diseases, diabetes, chronic kidney disease, other comorbidities, dyspnea, respiratory rate >20 breaths/min, lymphocyte count <$1.1\times10^9$/L, CD4+ T cells <550 cells/μL, CD8 + T cells <320 cells/μL, hemoglobin (Hb) <90 g/L, platelet count <$100\times10^9$/L, albumin <35g/L, direct bilirubin >8 μmol/L, aspartate aminotransferase (AST) >40 U/L, cystatin C >1.55 mg/L, CK-MB >8 U/L, brain natriuretic peptide (BNP) >500 pg/mL and >1000 pg/mL, CK >190 U/L, LDH >245 U/L, CRP >10 mg/L, ESR >20 mm/h, PCT >0.5 ng/L, IL-2 >10 pg/mL, IL-8>62 pg/mL, IL-10 >9 pg/mL, D-dimer >1.0 mg/mL, $PaO_2$ <80 and <60 mmHg, $PaO_2/FiO_2$ < 200 mmHg, CT consolidation, and CT score >10.

Before performing multivariate Cox regression analysis, we excluded CD4+ and CD8+ T cells, cystatin C, BNP, ESR, IL-2, IL-6, IL-8, IL-10, PaO2, and PaO2/FiO2 as they had more than 20% missing values (see details in Table 5); subsequently, the remaining variables that had statistical significance in univariate analysis were entered into the multivariate Cox regression analysis. We found that Hb<90 g/L, CK-MB >8 U/L, LDH > 245 U/L, PCT >0.5 ng/mL, and CT score >10 points were predictive of mortality (all $p<0.01$) (Table 6).

**Table 4. Comparison of treatment and complications of COVID-19 patients between the survivor and non-survivor groups.**

| Variable | Total (n = 432) | Non-survivors (n = 22) | Survivors (n = 410) | p value |
|---|---|---|---|---|
| Antiviral drugs | 417 (96.5) | 21 (95.5) | 396 (96.6) | 0.778 |
| Umifenovir | 237 (54.9) | 12 (54.5) | 225 (54.9) | 0.976 |
| Interferon ɑ nebulization | 265 (61.3) | 18 (81.8) | 247 (60.2) | 0.043 |
| Lopinavir/ritonavir | 257 (59.5) | 13 (59.1) | 244 (59.5) | 0.969 |
| Ribavirin | 29 (6.7) | 1 (4.5) | 28 (6.8) | 0.677 |
| Chloroquine | 25 (5.8) | 0 (0.0) | 25 (6.1) | 0.233 |
| Hydroxychloroquine | 28 (6.5) | 0 (0.0) | 28 (6.8) | 0.205 |
| Oselatmivir | 239 (55.3) | 17 (77.3) | 222 (54.1) | 0.034 |
| Antibiotics | 378 (87.5) | 21 (95.5) | 357 (87.1) | 0.247 |
| Corticosteroids | 106 (24.5) | 18 (81.8) | 88 (21.5) | <0.0001 |
| Intravenous immunoglobulin | 51 (11.8) | 9 (40.9) | 42 (10.2) | <0.0001 |
| Oxygen supply method | 359 (83.1) | 22 (100.0)) | 337 (82.2) | 0.03 |
| Nasal cannula | 299 (69.2) | 7 (31.8) | 292 (71.2) | <0.0001 |
| Nasal and mouth mask | 20 (4.6) | 2 (9.1) | 18 (4.4) | 0.271 |
| High-flow oxygen therapy | 5 (1.2) | 0 (0.0) | 5 (1.2) | 0.602 |
| Noninvasive mechanical ventilation | 26 (6.0) | 7 (31.8) | 19 (4.6) | <0.0001 |
| Invasive mechanical ventilation | 6 (1.4) | 6 (27.3) | 0 (0.0) | <0.0001 |
| Continuous renal replacement therapy | 9 (2.1) | 3 (13.6) | 6 (1.5) | 0.008 |
| Complications | 52 (12.0) | 19 (86.4) | 33 (8.0) | <0.0001 |
| ARDS | 24 (5.6) | 18 (81.8) | 6 (1.5) | <0.0001 |
| Acute kidney injury | 16 (3.7) | 3 (13.6) | 13 (3.2) | 0.042 |
| Heart failure | 8 (1.9) | 5 (22.7) | 3 (0.7) | <0.0001 |
| Acute hepatic insufficiency | 30 (6.9) | 6 (27.3) | 24 (5.9) | <0.0001 |
| Acute cardiac injury | 3 (0.7) | 2 (9.1) | 1 (0.2) | <0.0001 |
| Thrombocytopenia | 9 (2.1) | 5 (22.7) | 4 (1.0) | <0.0001 |
| Duration of viral shedding after COVID-19 onset, days | 6 (3–10) | 7 (4–7) | 6 (3–10) | 0.899 |

Abbreviations: COVID-19, coronavirus disease-2019; ARDS, acute respiratory distress syndrome.

## Characteristics of severe and non-severe COVID-19 patients

The differences in the demographic and clinical data, laboratory findings, CT image results, treatment approaches, and complications between the severe and non-severe COVID-19 patients are outlined in S1–S4 Tables.

## Discussion

The present study compared the demographic and clinical data, laboratory findings, radiological characteristics, and complications between surviving and non-surviving COVID-19 patients and evaluated the risk factors for mortality in two centers of Hubei province, China. The results showed that as compared with survivors, non-survivors were older; and they had a higher frequency of comorbidities, decreased lymphocyte count with lower T cell subsets, and Hb, platelet, and albumin levels. Biomarkers of inflammation, cytokines, liver and renal dysfunction, and cardiac and muscle injury were also markedly increased in non-surviving patients. A higher CT score accompanied by a higher rate of consolidation was found in dead patients. More non-surviving patients developed ARDS and died of respiratory failure. Multivariate Cox regression analysis demonstrated that low Hb (<90 g/L), high CK-MB (>8 U/L), LDH (>245 U/L), PCT (>0.5 ng/ml), and CT score (>10 points) were closely associated with the mortality in patients with COVID-19.

**Table 5. Univariate Cox regression analysis of factors associated with mortality.**

| Variables | Total | No-survivors | Survivors | Hazard risk (95% confidence interval) | *P* value |
|---|---|---|---|---|---|
| Age, years | | | | | |
| <65 | 308 (71.3) | 10 (45.5) | 298 (72.7) | 1 (ref) | 1 (ref) |
| ≥65 | 124 (28.7) | 12 (54.5) | 112 (27.3) | 2.730 (1.178–6.328) | 0.019 |
| Hypertension | 97 (22.5) | 13 (59.1) | 84 (20.5) | 4.609 (1.968–10.794) | <0.0001 |
| Cardiovascular or cerebrovascular diseases | 25 (5.8) | 7 (31.8) | 18 (4.4) | 9.285 (3.781–22.800) | <0.0001 |
| Diabetes | 56 (13.0) | 8 (36.4) | 48 (11.7) | 3.431 (1.437–8.193) | 0.005 |
| Chronic kidney disease | 9 (2.1) | 5 (22.7) | 4 (1.0) | 13.448 (4.950–36.535) | <0.0001 |
| Others | 25 (5.8) | 7 (31.8) | 18 (4.4) | 7.163 (2.918–17.585) | <0.0001 |
| Dyspnea | 35 (8.1) | 8 (36.4) | 27 (6.6) | 7.497 (3.138–17.912) | <0.0001 |
| Heart rate, beats/min | | | | | |
| ≤80 | 124 (28.7) | 3 (13.6) | 121 (29.5) | 1 (ref) | 1 (ref) |
| >80 | 308 (71.3) | 19 (86.4) | 289 (70.5) | 2.648 (0.784–8.951) | 0.117 |
| Respiratory rate, breaths/min | | | | | |
| ≤20 | 208 (48.1) | 4 (18.2) | 204 (49.8) | 1 (ref) | 1 (ref) |
| >20 | 224 (51.9) | 18 (81.8) | 206 (50.2) | 4.174 (1.413–12.336) | 0.010 |
| Lymphocyte count (<1.1×10⁹/L) | | | | | |
| ≥1.1 | 248 (57.4) | 3 (13.6) | 245 (59.8) | 1 (ref) | 1 (ref) |
| <1.1 | 184 (42.6) | 19 (86.4) | 165 (40.2) | 7.928 (2.346–26.800) | 0.001 |
| CD4+ T cells, cell/μL* | | | | | |
| ≥550 | 61 (58.1) | 0 (0.0) | 61 (67.8) | 1 (ref) | 1 (ref) |
| <550 | 44 (41.9) | 15 (100.0) | 29 (32.2) | 96.693 (1.364–6853.906) | 0.035 |
| CD8+ T cells, cells/μL* | | | | | |
| ≥320 | 58 (55.2) | 0 (0.0) | 58 (64.4) | 1 (ref) | 1 (ref) |
| <320 | 47 (44.8) | 15 (100.0) | 32 (35.6) | 88.356 (1.300–6004.710) | 0.037 |
| Hemoglobin, g/L | | | | | |
| ≥90 | 418 (96.8) | 18 (81.8) | 400 (97.6) | 1 (ref) | 1 (ref) |
| <90 | 14 (3.2) | 4 (18.2) | 10 (2.4) | 11.775 (3.946–35.143) | <0.0001 |
| Platelet count, ×10⁹/L | | | | | |
| ≥100 | 371 (85.9) | 13 (59.1) | 358 (87.3) | 1 (ref) | 1 (ref) |
| <100 | 61 (14.1) | 9 (40.9) | 52 (12.7) | 4.646 (1.985–10.874) | <0.0001 |
| Albumin, g/L | | | | | |
| ≥35 | 303 (70.1) | 6 (27.3) | 297 (72.4) | 1 (ref) | 1 (ref) |
| <35 | 129 (29.9) | 16 (72.7) | 113 (27.6) | 5.650 (2.209–14.450) | <0.0001 |
| Total bilirubin, μmol/L | | | | | |
| ≤26 | 419 (97.0) | 20 (90.9) | 399 (97.3) | 1 (ref) | 1 (ref) |
| >26 | 13 (3.0) | 2 (9.1) | 11 (2.7) | 3.624 (0.846–15.525) | 0.083 |
| Direct bilirubin, μmol/L | | | | | |
| ≤8 | 416 (96.3) | 19 (86.4) | 397 (96.8) | 1 (ref) | 1 (ref) |
| >8 | 16 (3.7) | 3 (13.6) | 13 (3.2) | 4.230 (1.251–14.301) | 0.020 |
| Aspartate aminotransferase, U/L | | | | | |
| ≤40 | 373 (86.3) | 12 (54.5) | 361 (88.0) | 1 (ref) | 1 (ref) |
| >40 | 59 (13.7) | 10 (45.5) | 49 (12.0) | 4.873 (2.104–11.289) | <0.0001 |
| Cystatin C, mg/L* | | | | | |
| ≤1.55 | 110 (87.3) | 10 (62.5) | 100 (99.9) | 1 (ref) | 1 (ref) |
| >1.55 | 16 (12.7) | 6 (37.5) | 10 (9.1) | 4.974 (1.806–13.703) | 0.002 |
| Creatine kinase-MB, U/L | | | | | |
| ≤8 | 159 (39.6) | 3 (13.6) | 156 (39.6) | 1 (ref) | 1 (ref) |

*(Continued)*

**Table 5.** (Continued)

| Variables | Total | No-survivors | Survivors | Hazard risk (95% confidence interval) | P value |
|---|---|---|---|---|---|
| >8 | 224 (58.9) | 19 (86.4) | 224 (58.9) | 4.868 (1.437–16.493) | 0.011 |
| Cardiac troponin I, pg/mL | | | | | |
| ≤35 | 132 (93.6) | 13 (81.3) | 119 (95.2) | 1 (ref) | 1 (ref) |
| <35 | 9 (6.4) | 3 (18.8) | 6 (4.8) | 3.056 (0.869–10.741) | 0.082 |
| Brain natriuretic peptide, pg/mL* | | | | | |
| ≤500 | 148 (82.2) | 7 (43.8) | 141 (86.0) | 1 (ref) | 1 (ref) |
| 500–1000 | 14 (7.8) | 5 (31.3) | 9 (5.5) | 9.367 (2.966–29.581) | <0.0001 |
| >1000 | 18 (10.0) | 4 (25.0) | 14 (8.5) | 6.552 (1.908–22.500) | <0.0001 |
| Creatine kinase, U/L | | | | | |
| ≤190 | 382 (92.3) | 16 (72.7) | 366 (93.4) | 1 (ref) | 1 (ref) |
| >190 | 32 (7.7) | 6 (27.3) | 26 (6.6) | 4.067 (1.590–10.403) | 0.003 |
| Lactate dehydrogenase, U/L | | | | | |
| ≤245 | 277 (67.9) | 6 (27.3) | 271 (70.2) | 1 (ref) | 1 (ref) |
| >245 | 131 (32.1) | 16 (72.7) | 115 (29.8) | 5.326 (2.083–13.619) | <0.0001 |
| C-reactive protein, mg/L | | | | | |
| ≤10 | 185 (44.3) | 4 (18.2) | 181 (45.7) | 1 (ref) | 1 (ref) |
| >10 | 233 (55.7) | 18 (81.8) | 215 (54.3) | 3.650 (1.235–10.784) | 0.019 |
| Erythrocyte sedimentation rate, mm/h* | | | | | |
| ≤20 | 119 (39.5) | 2 (10.0) | 117 (41.6) | 1 (ref) | 1 (ref) |
| >20 | 182 (60.5) | 18 (90.0) | 164 (58.4) | 5.103 (1.83–22.013) | 0.029 |
| Procalcitonin, ng/mL | | | | | |
| ≤0.5 | 373 (94.7) | 19 (86.4) | 354 (95.2) | 1 (ref) | 1 (ref) |
| >0.5 | 21 (5.3) | 3 (13.6) | 18 (4.8) | 3.711 (1.094–12.592) | 0.035 |
| Interleukin-2 receptor, U/mL* | | | | | |
| ≤710 | 84 (76.4) | 2 (22.2) | 82 (81.2) | 1 (ref) | 1 (ref) |
| >710 | 26 (23.6) | 7 (77.8) | 19 (18.8) | 14.062 (2.893–68.359) | 0.001 |
| Interleukin-6, pg/mL* | | | | | |
| ≤7 | 77 (70.0) | 0 (0.0) | 77 (76.2) | 1 (ref) | 1 (ref) |
| >7 | 33 (30.0) | 9 (100.0) | 24 (23.8) | 402.610 (0.198–816985.268) | 0.123 |
| Interleukin-8, pg/mL* | | | | | |
| ≤62 | 107 (97.3) | 7 (77.8) | 100 (90.0) | 1 (ref) | 1 (ref) |
| >62 | 3 (2.7) | 2 (22.2) | 1 (1.0) | 13.434 (2.762–65.339) | 0.001 |
| Interleukin-10, pg/mL* | | | | | |
| ≤9 | 103 (93.6) | 7 (77.8) | 96 (95.0) | 1 (ref) | 1 (ref) |
| >9 | 7 (6.4) | 2 (22.2) | 5 (5.0) | 5.028 (1.025–24.674) | 0.047 |
| D-dimer, μg/ml | | | | | |
| ≤0.5 | 168 (39.9) | 3 (13.6) | 165 (41.4) | 1 (ref) | 1 (ref) |
| 0.5–1.0 | 178 (42.3) | 7 (31.8) | 171 (42.9) | 2.78 (0.563–8.428) | 0.259 |
| >1.0 | 75 (17.8) | 12 (54.5) | 63 (15.8) | 9.825 (2.770–34.853) | <0.0001 |
| PaO$_2$, mmHg* | | | | | |
| ≥80 | 149 (70.0) | 5 (22.7) | 144 (75.4) | 1 (ref) | 1 (ref) |
| 60–79.9 | 50 (23.5) | 11 (50.0) | 39 (20.4) | 6.456 (2.243–18.580) | 0.001 |
| <60 | 14 (6.6) | 6 (27.3) | 8 (4.2) | 13.916 (4.245–45.621) | <0.0001 |
| PaO$_2$/FiO$_2$, mmHg* | | | | | |
| ≥200 | 114 (53.5) | 0 (0.0) | 114 (59.7) | 1 (ref) | 1 (ref) |
| <200 | 99 (46.5) | 22 (100.0) | 77 (40.3) | 72.483 (2.333–2251.514) | 0.015 |
| CT consolidation | 97 (23.3) | 10 (45.5) | 87 (22.1) | 2.817 (1.217–6.521) | 0.016 |

(*Continued*)

**Table 5.** (Continued)

| Variables | Total | No-survivors | Survivors | Hazard risk (95% confidence interval) | P value |
|---|---|---|---|---|---|
| CT score | | | | | |
| ≤10 | 369 (88.7) | 5 (22.7) | 364 (92.4) | 1 (ref) | 1 (ref) |
| >10 | 47 (11.3) | 17 (77.3) | 30 (7.6) | 29.171 (10.759–79.092) | <0.0001 |

*: More than 20% missing values existed in this variable.

ref: reference.

At the early stage in 2020, the mortality rate of COVID-19 was nearly 3.7–5.4% [5–7]. The mortality rate increased up to 52.4% in patients suffering from ARDS [9] and 61.5% among severe and critical cases [8]. A total of 5.09% of patients died among our study population at two centers in Hubei province, which was consistent with previous studies from China. The mortality rate increased to 17.6% among patients with severe and critical types (S1 Table), which was significantly lower than the early published data. We suspected that this discrepancy was related to the relatively abundant treatment experience during the later period of this pandemic. Our data showed that non-surviving patients were older and had underlying diseases, which was similar to that in most of the published studies [6, 9, 12, 13, 16, 17]. As reported in previous studies presented [3], the top 3 common symptoms were fever, cough, and fatigue. There were no differences in gender, symptoms except dyspnea, and illness onset to admission between the survivor and non-survivor groups [11]. Contradictory to one previous study [10], we failed to find any difference in angiotensin-converting enzyme inhibitor (ACEI) or angiotensin receptor blocker (ARB) drug administration between the two groups.

SARS-CoV-2 can cause systemic multiple organ dysfunction mainly in the lungs by binding to the angiotensin-converting enzyme 2 (ACE2) receptors [4], resulting in cytokine storm and immune damage. In the present study, several aberrant biomarkers that represent organ dysfunction were detected. As compared with surviving patients, non-surviving patients had lower lymphocyte count. The subsets of lymphocytes, and CD4+ and CD8+ T cells, were also dramatically decreased in non-survivors. As the biomarkers of cytokine storm, levels of most of the ILs were increased among COVID patients. The levels of cytokines were significantly higher in non-survivors as compared with survivors. The levels of inflammatory biomarkers, including CRP, ESR, and PCT, were also markedly higher in non-survivors. Furthermore, we observed that high levels of PCT (>0.5 ng/mL) were an important poor prognostic predictor; thus, confirming the results of published data [6, 13]. The results of the above-mentioned

**Table 6. Multivariate Cox regression analysis of factors associated with mortality.**

| Variables | Hazard Risk (95% confidence interval) | P value |
|---|---|---|
| Hemoglobin <90 g/L | 10.776 (3.075–37.766) | <0.0001 |
| Creatine kinase-MB >8 U/L | 9.155 (2.424–34.584) | 0.001 |
| Lactate dehydrogenase >245 U/L | 5.963 (2.029–17.529) | 0.001 |
| Procalcitonin >0.5 ng/mL | 7.080 (1.671–29.992) | 0.008 |
| CT score >10 | 39.503 (12.430–125.539) | <0.0001 |

Except for variables, such as CD4+ and CD8+ T cells, cystatin C, brain natriuretic peptide, erythrocyte sedimentation rate, interleukin-2, -6, -8, -10, $PaO_2$, and $PaO_2/FiO_2$ that had more than 20% missing values (see details in Table 5), all variables that were statistically different in univariate Cox regression analysis were entered into multivariate analysis.

biomarkers representing cytokine storm and inflammation were consistent with those in several previous studies [11, 13, 18]. Abnormal laboratory findings in non-survivors included increased total and direct bilirubin, AST, cystatin C, cardiac troponin I, CK-MB, BNP, and D-dimer, indicating aberrant hepatic damage, kidney insufficiency, myocardial injury, and aberrant coagulation, respectively. Although most of the biochemical variables were not entered into the multivariate Cox regression model, CK-MB >8 U/L was one of the predictors of mortality, suggesting that COVID-19 patients who develop acute cardiac injury are prone to have a poor prognosis [13, 16, 18]. In our study, hypoproteinemia and anemia were more frequently detected in patients who did not survive [5]. Moreover, multivariate Cox regression analysis showed that moderate anemia (Hb <90 g/L) was an independent risk factor of mortality. We also found that there were increased LDH and creatine kinase levels in dead and severe cases. LDH (>245 U/L) was also found to be a predictor of mortality of COVID-19 patients. The results confirmed the previous conclusion that LDH and creatine kinase can be prognostic biomarkers in COVID-19 patients [12, 19]. We observed that decreased $PaO_2$ and $PaO_2$: $FiO_2$ were detected in non-surviving COVID-19 patients, suggesting that SARS-CoV-2 mainly caused acute lung injury, especially in severe cases.

In accordance with most of the recent studies [3, 10], the most common CT features of COVID-19 included bilateral ground-glass opacity and consolidation. High rate of consolidation was found in non-surviving patients. Furthermore, we evaluated the CT features with a semi-quantitative score [20, 21]. Undoubtedly, an increased CT score was detected in dead patients. Multivariate Cox regression showed that CT score >10 was an independent risk factor of mortality, indicating that more lung lobes were involved in non-survivors. This finding was consistent with previous research results [3, 10]. Pleural effusion and pleural thickening were less frequently observed in our current study.

No effective therapeutic approach against SARS-CoV-2 has been confirmed. In this retrospective analysis study, 96.5% of patients received antiviral treatment. The antiviral drugs included umifenovir, interferon ⟨ nebulization, lopinavir/ritonavir, ribavirin, chloroquine, hydroxychloroquine, and oseltamivir. No significant difference in antiviral drug usage was detected between survivors and non-survivors. Our results were consistent with those of previous studies [10, 17]; thus, confirming that there is no effective drug against SARS-CoV-2. Regarding antibiotic use, we found that no significant difference between survivors and non-survivors. However, the overall rate of using antibiotics in this study and previous other studies was significantly high, indicating that the clinician was more prone to combining antiviral and antibiotic therapies in the current pandemic, especially in those with severe and critical types.

Cytokine storm and systemic inflammation were considered as the novel physiological features of COVID-19. Some therapeutic approaches were attempted to attenuate these features. Evidence has shown that corticosteroids can reduce overreaction of inflammation and cytokine storm; low dose (1–2 mg/kg) of corticosteroids for 3–5 days is recommended in guideline [14]. But the disadvantages are also obvious as they can inhibit the immune function; thus, increasing the chances of secondary infection and prolonging the viral shedding period. There were more number of patients receiving corticosteroid treatment among non-survivors in our study. In addition, intravenous immunoglobulin was also commonly used in non-surviving patients. Our results demonstrated that a high frequency of systemic inflammation and cytokine storm occurred in deceased patients. Clinicians tried using corticosteroids and immunoglobulin to reverse these conditions, but all of these therapies were ineffective.

Yang X et al. [8] demonstrated that 81.0% of non-survivors developed ARDS among the critically ill COVID-19 patients. In accordance with previous studies [8, 9, 22], we found that ARDS (81.8%) was the topmost cause of death. It has been shown that SARS-CoV-2 binds to the ACE2 receptor to invade host cells, especially in the lungs, kidney, heart, contributing to

cytokine storm and inflammatory state, both of which are implicated in multi-organ dysfunction [23]. Acute cardiac injury, acute hepatic insufficiency [17], acute kidney injury, and thrombocytopenia were also commonly detected in deceased COVID-19 patients. A systemic review and meta-analysis illustrated that acute cardiac injury and acute kidney injury are tightly associated with an increased risk of COVID-19 related mortality [16]. A study by Cao J et al. [17] showed 13 cases of acute liver injury in 17 non-survivors. Based on the results of our study and previous studies, we confirmed that multi-organ dysfunction involving the lung, heart, liver, kidney, and coagulation caused by SARS-CoV-2 [24], was more frequently observed in the deceased patients.

Several limitations of our study should be mentioned. First, since this was a retrospective study at two centers of Hubei province, all laboratory tests were not performed on all of the patients. Therefore, the effect of missing variables might have been underestimated in the prediction of mortality. Second, the insufficiency of viral RNA detection may have resulted in inaccuracy of viral shedding. Finally, interpretation of our results might have been limited by the small sample size in the non-survivor group.

In conclusion, our study found that COVID-19 patients who did not survive were old and they had more underlying diseases. Several aberrant laboratory findings, which indicated cytokine storm, inflammation, acute cardiac injury, acute hepatic damage, and acute renal insufficiency, were also detected in non-surviving patients. Death in most of the deceased patients resulted from ARDS. Cox regression analysis was conducted to identify the following 5 predictors: Hb <90 g/L, CK-MB >8 U/L, LDH >245 U/L, PCT >0.5 ng/ml, and CT score >10, of increased mortality among the overall population of COVID-19 patients. The results of our study confirmed the previous findings, and they highlighted early biomarkers for the risk of mortality in patients with COVID-19. Furthermore, our study provided evidence for early intervention and reasonable allocation of medical resources in this global pandemic.

## Supporting information

**S1 Table. Demography and clinical characteristics of different severity of COVID-19 patients.**
(DOCX)

**S2 Table. Laboratory findings of COVID-19 patients between survivors and non-survivors.**
(DOCX)

**S3 Table. CT image results of different severity of COVID-19 patients***.
(DOCX)

**S4 Table. Treatment and complications between severe and non-severe COVID-19 patients.**
(DOCX)

**S5 Table. Normal range of laboratory findings of COVID-19 patients.**
(DOCX)

## Acknowledgments

### Declarations

We sincerely appreciate all front-line medical staff for their hard work and sacrifice.

## Author Contributions

**Conceptualization:** Xiao-Bin Zhang, Li-Da Chen, Hui-Qing Zeng.

**Data curation:** Lan Hu, Qiu-Fen Huang, Yu-Qing Cai, Hui-Qing Zeng.

**Formal analysis:** Quan Ming.

**Funding acquisition:** Xiao-Bin Zhang.

**Investigation:** Xiao-Bin Zhang, Lan Hu, Quan Ming, Xiao-Jie Wei, Zhen-Yu Zhang, Ming-Hui Wang, Weng-Zhen Yao.

**Methodology:** Lan Hu, Xiao-Jie Wei, Zhen-Yu Zhang, Li-Da Chen, Ming-Hui Wang, Weng-Zhen Yao, Zhang-Qiang Ye, Yu-Qing Cai, Hui-Qing Zeng.

**Project administration:** Li-Da Chen, Zhang-Qiang Ye.

**Software:** Zhen-Yu Zhang, Weng-Zhen Yao.

**Supervision:** Xiao-Bin Zhang, Hui-Qing Zeng.

**Validation:** Xiao-Jie Wei, Yu-Qing Cai.

**Visualization:** Qiu-Fen Huang.

**Writing – original draft:** Xiao-Bin Zhang, Quan Ming, Ming-Hui Wang, Qiu-Fen Huang, Zhang-Qiang Ye, Hui-Qing Zeng.

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
