## [Decision Letter · Decision Letter 0]

21 Dec 2020

PONE-D-20-31483

Risk factors for mortality of coronavirus disease-2019 (COVID-19) patients in two centers of Hubei province, China: a retrospective analysis

PLOS ONE

Dear Dr. Zhang

Thank you for submitting your manuscript to PLOS ONE. After careful consideration, we feel that it has merit but does not fully meet PLOS ONE’s publication criteria as it currently stands. Therefore, we invite you to submit a revised version of the manuscript that addresses the points raised during the review process.

The following reviews were submitted by external reviewers and the consensus was that this manuscript is requires minor revisions. Please respond ti the reviewers' comments as noted below and submit a revised manuscript in the form as outlined buy PLoS One editorial office. We look forward to receiving your revised manuscript. 

Reviewer 1 - ACADEMIC EDITOR: Love the paper and like the information but could really benefit from a native English speaker edit

----

Line 27--We aim to is awkward-- We analyzed

Line 43- Use standard abbreviation for hemoglobin (Hb) of (Hgb). 

Line 65- I think you should just change dissimilarity to homology. I thin you misinterpreted the original 'Notably, 2019-nCoV was closely related (with 88% identity) to two bat-derived severe acute respiratory syndrome (SARS)-like coronaviruses, bat-SL-CoVZC45 and bat-SL-CoVZXC21, collected in 2018 in Zhoushan, eastern China, but were more distant from SARS-CoV (about 79%) and MERS-CoV (about 50%).'

Line 80- the tense is odd..Instead of We tried. ...We provide

these things continue but as above-just a good edit

Line 118. is it CK or CK-MB?  Would also include normal ranges for all measurements

Reviewer 2 Comments

Zhang et al. report clinical characteristics of 432 COVID-19 patients from two hospitals in Wuhan, China. The authors stratify their analyses by survivors (n=410) and non-survivors(n=22). Using multivariate Cox regression analyses, the authors determined that hemoglobin, creatine kinase-MB, lactate dehydrogenase and procalcitonin levels are predictors of COVID-19 mortality. This is consistent with similar published reports. Small number of non-survivors group is a limitation of the study, not acknowledged in discussion. Nonetheless, the data on the survivors group is informative and worth publishing. Typically, predictive models using the identified risk factors are presented in these studies (reviewer does not think this is needed for acceptance, but that would be helpful). Authors should address the following:

1. Please provide details on laboratory testing methodologies (lines 115-122).

2. Please clarify “ref” values in Table 5.

We look forward to receiving your revised manuscript.

Kind regards,

Alexandra Lucas

Academic Editor

PLOS ONE

Journal Requirements:

" The funders had no role in study design, data collection and analysis, decision to publish, or preparation of the manuscript."

"This work was supported by Grant 2018-2-65 for Youth Research Fund

 from Fujian Provincial Health Bureau, and Grant 2018J01393 for Fund from Natural

Science Foundation of Fujian Province, China."

Reviewers' comments:

Reviewer's Responses to Questions

**Comments to the Author**

1. Is the manuscript technically sound, and do the data support the conclusions?

Reviewer #1: Yes

Reviewer #2: Yes

2. Has the statistical analysis been performed appropriately and rigorously? 

Reviewer #1: Yes

Reviewer #2: Yes

3. Have the authors made all data underlying the findings in their manuscript fully available?

Reviewer #1: Yes

Reviewer #2: Yes

4. Is the manuscript presented in an intelligible fashion and written in standard English?

Reviewer #1: Yes

Reviewer #2: Yes

5. Review Comments to the Author

Reviewer #1: Love the paper and like the information but could really benefit from a native English speaker edit

----

Line 27--We aim to is awkward-- We analyzed

Line 43- Use standard abbreviation for hemoglobin (Hb) of (Hgb).

Line 65- I think you should just change dissimilarity to homology. I thin you misinterpreted the original 'Notably, 2019-nCoV was closely related (with 88% identity) to two bat-derived severe acute respiratory syndrome (SARS)-like coronaviruses, bat-SL-CoVZC45 and bat-SL-CoVZXC21, collected in 2018 in Zhoushan, eastern China, but were more distant from SARS-CoV (about 79%) and MERS-CoV (about 50%).'

Line 80- the tense is odd..Instead of We tried. ...We provide

these things continue but as above-just a good edit

Line 118. is it CK or CK-MB? Would also include normal ranges for all measurements

Reviewer #2: Zhang et al. report clinical characteristics of 432 COVID-19 patients from two hospitals in Wuhan, China. The authors stratify their analyses by survivors (n=410) and non-survivors(n=22). Using multivariate Cox regression analyses, the authors determined that hemoglobin, creatine kinase-MB, lactate dehydrogenase and procalcitonin levels are predictors of COVID-19 mortality. This is consistent with similar published reports. Small number of non-survivors group is a limitation of the study, not acknowledged in discussion. Nonetheless, the data on the survivors group is informative and worth publishing. Typically, predictive models using the identified risk factors are presented in these studies (reviewer does not think this is needed for acceptance, but that would be helpful). Authors should address the following:

1. Please provide details on laboratory testing methodologies (lines 115-122).

2. Please clarify “ref” values in Table 5.

6. PLOS authors have the option to publish the peer review history of their article (what does this mean?). If published, this will include your full peer review and any attached files.

Reviewer #1: **Yes: **Daniel O Griffin

Reviewer #2: No

---

## [Author Response · Author response to Decision Letter 0]

7 Jan 2021

Department of Pulmonary and Critical Care Medicine, Zhongshan Hospital, Xiamen University; 

Teaching Hospital of Fujian Medical University; 

No.201, Siming District, Xiamen, Fujian Province, 361004, People's Republic of China.

Jan,12th, 2021

Re: Manuscript No. PONE-D-20-31483

Title: Risk factors for mortality of coronavirus disease-2019 (COVID-19) patients in two centers of Hubei province, China: a retrospective analysis

Dear Editor Alexandra Lucas, 

Thank you for the timely review of our manuscript. 

We have attached our point-by-point responses to the reviewers’ and editors’ suggestions and have also responded to the reviewers’ comments in the text by using the ‘track changes’ function in the ‘revised revision’. 

This manuscript has been edited and proofread by Medjaden Bioscience Limited.

We hope that the revised manuscript is now acceptable for publication in your journal.

I look forward to hearing from you soon.

Best regards!

Sincerely yours,

Corresponding author: Xiao-Bin Zhang 

Email: zhangxiaobincn@xmu.edu.cn

Response to the reviewers and editor's comments

Response to the reviewers' comments

Reviewer 1 - ACADEMIC EDITOR:

Comment 1: Love the paper and like the information but could really benefit from a native English speaker edit

Response 1: Thank you for your kind comment. Our manuscript has been edited by a native English speaker editor working in a language editing company (Medjaden Incorporate, www.medjaden.com). The language certificate is attached for your reference.

Comment 2: Line 27--We aim to is awkward-- We analyzed

Response 2: Thank you for your valuable suggestion. We have re-written this sentence according to your suggestion.

Comment 3: Line 43- Use standard abbreviation for hemoglobin (Hb) of (Hgb). 

Response 3: Thank you for your valuable suggestion. The abbreviation for hemoglobin has been changed to Hb in the whole manuscript.

Comment 4: Line 65- I think you should just change dissimilarity to homology. I thin you misinterpreted the original 'Notably, 2019-nCoV was closely related (with 88% identity) to two bat-derived severe acute respiratory syndrome (SARS)-like coronaviruses, bat-SL-CoVZC45 and bat-SL-CoVZXC21, collected in 2018 in Zhoushan, eastern China, but were more distant from SARS-CoV (about 79%) and MERS-CoV (about 50%).'

Response 4: Thank you for your valuable comment. We apologized for our mistake. The relevant context has been corrected according to your comment as follows: “Being a beta coronavirus, the SARS-CoV-2 virus was closely related (with 88% identity) to two bat-derived severe acute respiratory syndrome (SARS)-like coronaviruses, bat-SL-CoVZC45 and bat-SL-CoVZXC21, collected in 2018 in Zhoushan, eastern China, but were more distant from SARS-CoV (about 79%) and MERS-CoV (about 50%)”.

Comment 5: Line 80- the tense is odd..Instead of We tried. ...We provide these things continue but as above-just a good edit

Response 5: Thank you for your valuable suggestion. This manuscript has been edited and proofread by Medjaden Bioscience Limited to ensure that it is readable. 

Comment 6: Line 118. is it CK or CK-MB? Would also include normal ranges for all measurements

Response 6: Thank you for your thoughtful question and valuable suggestion. Both CK and CK-MB were analyzed in our manuscript. The normal ranges for all measurements are outlined in eTable 5.

Reviewer 2 Comments

Comment 1: Zhang et al. report clinical characteristics of 432 COVID-19 patients from two hospitals in Wuhan, China. The authors stratify their analyses by survivors (n=410) and non-survivors(n=22). Using multivariate Cox regression analyses, the authors determined that hemoglobin, creatine kinase-MB, lactate dehydrogenase and procalcitonin levels are predictors of COVID-19 mortality. This is consistent with similar published reports. Small number of non-survivors group is a limitation of the study, not acknowledged in discussion. Nonetheless, the data on the survivors group is informative and worth publishing. Typically, predictive models using the identified risk factors are presented in these studies (reviewer does not think this is needed for acceptance, but that would be helpful). 

Response 1: Thank you for your positive feedback. We have acknowledged that our manuscript has several limitations, such as retrospective design, lack of several biochemical markers, and small number of patients in the non-survivor group. The small number of patients in the non-survivor group of our study has been included as a limitation in the Discussion section of the revised manuscript. Since we analyzed the predictive factors of survival in COVID-19 patients, Cox regression was considered the first choice (J Allergy Clin Immunol. 2020 Jul;146(1):110-118). 

Authors should address the following:

Comment 2. Please provide details on laboratory testing methodologies (lines 115-122).

Response 2: Thank you for your valuable suggestion. The methodologies of all laboratory tests have been presented in the “Methods” section of the revised manuscript.

Comment 3. Please clarify “ref” values in Table 5.

Response 3: Thank you for your thoughtful comment. The “ref” value in Table 5 is 1.0 (Lancet 2020; 395:1054-1062). This issue has been corrected in the revised manuscript. 

Response to the editor's comments

Comment 1. Please ensure that your manuscript meets PLOS ONE's style requirements, including those for file naming. The PLOS ONE style templates can be found at

 Response 1: Thank you for your valuable comment. The manuscript has been modified according to the journal style.

Comment 2. Thank you for stating the following in the Acknowledgments Section of your manuscript:

" The funders had no role in study design, data collection and analysis, decision to publish, or preparation of the manuscript."

"This work was supported by Grant 2018-2-65 for Youth Research Fund

 from Fujian Provincial Health Bureau, and Grant 2018J01393 for Fund from Natural

Science Foundation of Fujian Province, China."

 Response 2: Thank you for your valuable suggestion. The Funding information has been deleted from the Acknowledgements section and it has not been presented in any other sections of our manuscript. Meanwhile, we have included the Funding information in the cover letter as follows: “This work was supported by Grant 2018-2-65 for Youth Research Fund from Fujian Provincial Health Bureau, Grant 2020GGB057 for Young people training project from Fujian Province Health Bureau, and Grant 2018J01393 for Fund from Natural Science Foundation of Fujian Province, China.” 

Comment 3. Your ethics statement should only appear in the Methods section of your manuscript. If your ethics statement is written in any section besides the Methods, please move it to the Methods section and delete it from any other section. Please ensure that your ethics statement is included in your manuscript, as the ethics statement entered into the online submission form will not be published alongside your manuscript.

Response 3: Thank you for your valuable suggestion. Our ethics statement only appears in the “Methods” section of the revised manuscript.

We would like to take this opportunity to express our gratitude and appreciation for the meticulous and professional comments provided by the reviewers and editors that have contributed greatly to improving our work.

---

## [Editor Report · Decision Letter 1]

13 Jan 2021

Risk factors for mortality of coronavirus disease-2019 (COVID-19) patients in two centers of Hubei province, China: a retrospective analysis

PONE-D-20-31483R1

Dear Dr. Zhang,

We’re pleased to inform you that your manuscript has been judged scientifically suitable for publication and will be formally accepted for publication once it meets all outstanding technical requirements.

Kind regards,

Alexandra Lucas

Academic Editor

PLOS ONE
---

## [Editor Report · Acceptance letter]

18 Jan 2021

PONE-D-20-31483R1 

Risk factors for mortality of coronavirus disease-2019 (COVID-19) patients in two centers of Hubei province, China: a retrospective analysis 

Dear Dr. Zhang:

I'm pleased to inform you that your manuscript has been deemed suitable for publication in PLOS ONE. Congratulations! Your manuscript is now with our production department. 

Kind regards, 

on behalf of

Professor Alexandra Lucas 

Academic Editor

PLOS ONE